# Head to head comparison of two commercial fecal calprotectin kits as predictor of Mayo endoscopic sub-score and mucosal TNF expression in ulcerative colitis

Rasmus Goll [1,2]*, Richard Heitmann[3], Øystein Kittel Moe[2], Katrine Carlsen[4], Jon Florholmen[1,2]

**1** Institute of Clinical Medicine, UiT the Arctic University of Norway, Tromsø, Norway, **2** Medical Gastroenterology, Division of Internal medicine, University Hospital of North Norway, Tromsø, Norway, **3** Department of Gastroenterology, Telemark Hospital, Skien, Norway, **4** Department of Pediatrics, Hvidovre University Hospital, Copenhagen, Denmark

\* rasmus.goll@uit.no

**Data Availability Statement:** All relevant data are within the manuscript and its Supporting Information files.

## Abstract

### Background

Fecal calprotectin is widely used to monitor disease activity in patients with inflammatory bowel disease. Multiple commercial kits exist, however, since the analyses are not standardized, these kits cannot be used interchangeably. We aimed to perform a technical evaluation of two kits (Calpro from Calprolab, Norway and Calprest from Eurospital, Italy) and perform a tuning for detection of clinically relevant disease states in ulcerative colitis.

### Materials and methods

For tuning against different clinical states a total of 116 patients with ulcerative colitis were recruited (67 of which were part of an earlier publication). For the technical evaluation an additional series of 80 random samples from the hospital lab were included. Technical evaluation was done by correlation and limits of agreement analysis; cut-off levels were explored by ROC analysis against clinically relevant actual states.

### Results

The technical evaluation revealed good correlation between assays, however a non-linear difference was found: At values below 200 mg/kg, no significant bias was found; in the interval 200–1000 mg/kg the Calprest assay measured on average 30% lower than Calpro; and at higher values Calprest measured 60% higher values than Calpro. Both assays predicted Mayo endoscopic score (MES) 0 (cutoff 28: sensitivity 0.38; specificity 0.82 for Calprest; cut-off 28: sensitivity 0.50; specificity 0.77 for Calpro), and MES 2–3 (cutoff 148: sensitivity 0.72; specificity 0.80 for Calprest; cutoff 208: sensitivity 0.64; specificity 0.80 for Calpro), but did not predict normalization of mucosal TNF transcript *per se*. A combination of calprotectin

**Funding:** The publication charges for this article have been funded by a grant from the publication fund of UiT The Arctic University of Norway. The funder had no role in study design, data collection and analysis, decision to publish, or preparation of the manuscript.

**Competing interests:** The authors have declared that no competing interests exist.

and MES predicted mucosal TNF transcript values reasonably well (Calpro: sensitivity 0.85, specificity 0.58; Calprest: sensitivity 0.85, specificity 0.61).

## Conclusion

The Calpro and Calprest assays correlated well, but subtle differences were found, underlining the need for kit-specific cut-off values. Both kits were most precise in predicting active inflammation (MES 2–3), but less so for prediction of mucosal healing (MES 0) and normalization of mucosal TNF gene expression.

## Introduction

Assessment of disease activity in ulcerative colitis (UC) pose several problems as there is often a discordance between patient reported symptom burden and actual state of the colon mucosa [1]. Fecal calprotectin has become an important non-invasive biomarker for surveillance of inflammatory activity in inflammatory bowel disease. To some extent, the daily clinical decisions can be made based on this analysis [2]. The overall target for the treatment of UC should be resolution of inflammation as mounting evidence indicate improved prognosis for patients who achieve deep remission (i.e. healing of mucosa) [3,4]. The state of deep remission decreases risk of long term complications and necessity for surgery [5]. In clinical practice the finding of Mayo endoscopic subscore (MES) 0–1 is considered "endoscopic remission" [3]. Recent data, however, indicates that MES 1 is not completely normal with regards to inflammatory histological state [6]. This increases the risk of relapse [7] and later complications. To achieve complete inflammatory remission, the aim should be MES 0. Another method to evaluate the activation state of the mucosal immune system is to quantify the Tumor Necrosis Factor (TNF) gene activity by tissue qPCR [8] from colon biopsies.

Several publications have aimed to assess fecal calprotectin measurements towards prediction of endoscopic findings [9–11]. There are however discrepancies between recommended cut-off values, which probably depend on technical differences between kits, as the method is not standardized. The availability of different commercial kits raises concern that cut-off values are not universal, but kit-specific. The focus of the present paper is to compare the performance of two commercial calprotektin ELISA kits: Calpro (Calprolab, Norway) and Calprest (Eurospital, Italy), and perform a tuning for detection of clinically relevant states of UC.

## Materials and methods

Approval by the North Norwegian Regional Committee of Medical Ethics was granted (2012/1349). All study participants received information of the study and signed a consent form. Sixty-six participants with UC were recruited for an earlier publication [12] as part of a prospective project at the Department of Gastroenterology, University hospital of North Norway; additionally, 50 subjects with UC and 16 normal controls were recruited for the present study (please see Fig 1 and Table 1 for details). The participants were asked to deliver two separate fecal samples taken on different days during the week before endoscopy. At endoscopy participants were scored according to MES and samples for tissue qPCR were collected.

Due to missing values the actual number of observations in each analysis vary (Table 1). For day-to-day variation, dual samples were available from 51 individuals, obtained within 1

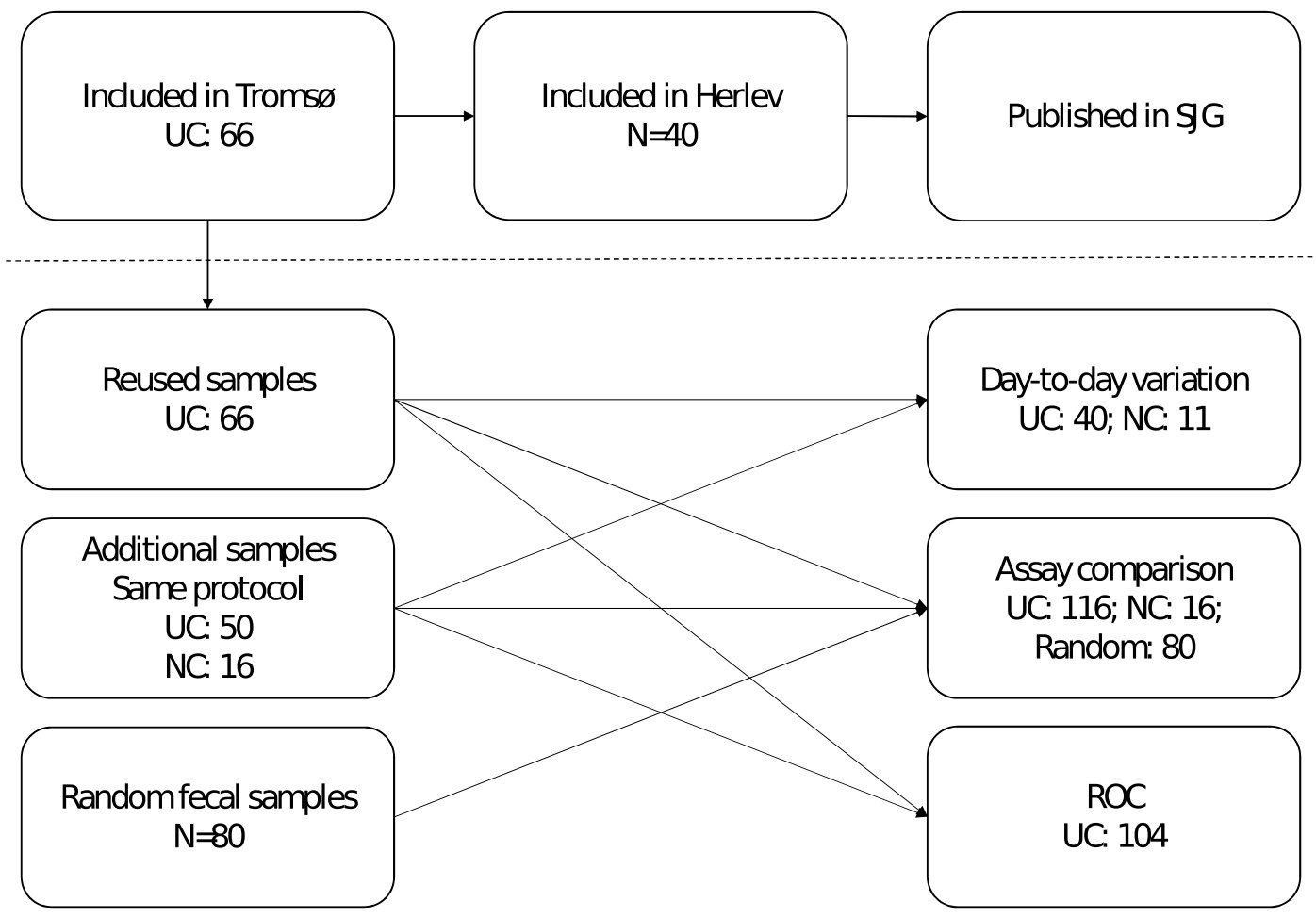

**Fig 1. Overview of available samples in the study.** UC: patients with ulcerative colitis; NC: normal controls. Abbreviations: UC: Ulcerative Colitis; NC: Normal Controls; ROC: Receiver operating characteristic; SJG; Scandinavian Journal of Gastroenterology.

week (43 of these were part of the previous publication [12]). These dual samples were measured on both kits, and the variation between results analysed.

For head to head comparison of the two ELISA assays, the UC and normal control samples mentioned above were used. In addition, 80 samples from the routine calprotectin workflow at the hospital lab were picked at random, extracted, and analysed on both assays. These samples were only used for technical evaluation; thus no clinical information was obtained. For the most part those samples would be from IBD patients (both UC and Crohn's disease), or samples used for diagnostic work-up in cases of chronic diarrhoea.

For 104 of the UC samples, MES were also available, which were used for ROC curves (67 of these observations were part of a previously published paper [12]). For 96 patients, a combination of TNF and calprotectin and MES values were available.

Both kits contain extraction devices for accurate sampling of the desired amount of fecal matter. The samples were mixed and extracted according to the manufacturer's instructions for Calpro (Calprolab, Norway) and Calprest (Eurospital, Italy). The extracts were analysed on an automated system (Dynex DS2 ELISA processor, Dynex, Germany).

**Table 1. Available samples and key metadata in the different analyses.**

| Series | | Ulcerative colitis | | | | Normal control | Total N |
|---|---|---|---|---|---|---|---|
| **Day to day variation*** | N | 8 | 18 | 10 | 4 | 11 | 51* |
| | MES | 0 | 1 | 2 | 3 | 0 | |
| **Assay comparison** | N | 43 | 42 | 43 | 42 | 16 | 186** |
| | | Quartile 1 | Quartile 2 | Quartile 3 | Quartile 4 | Mean (Min–max) | |
| | Calpro | 0–25 | 26–137 | 138–595 | 596–1485 | 64 (24–195) | |
| | Calprest | 0–35 | 36–120 | 121–386 | 387–3000 | 52 (24–145) | |
| **ROC** | n | 104** | 104** | 96 | 96 | | 104 |
| | Analysis | MES = 0 by Calprotectin | MES ≥2 by Calprotectin | TNF by Calprotectin | TNF by Calprotectin+MES | | |
| | Pos/neg | 42/62 | 25/79 | 27/69 | 27/69 | | |

*43 and

**66 of these samples were used in an earlier publication [12]. Calprotectin values are mg/kg. MES: Mayo endoscopic sub-score. ROC: receiver operating characteristics. TNF: Tumor necrosis factor gene expression.

The method for TNF measurements have been published earlier [13]–in brief: biopsies were immediately immersed in RNA-later (Qiagen, Hilden, Germany) for storage. The biopsies were then disintegrated using a Magnalyzer (Roche, Germany) ceramic bead shaker system and RNA was isolated using the Allprep DNA/RNA Mini Kit (Qiagen) on a Qiacube instrument (Qiagen). Quantity and purity of the extracted RNA were determined using the Qubit 3 Fluorometer (Thermo Fisher Scientific, Waltham, MA, USA). cDNA was then produced with QuantiTect Reverse Transcription Kit (Qiagen), and qPCR was run using TaqMan assays for TNF with ACTB as housekeeping gene (previously published [13]; primers/probes ordered from Eurogentec, Liège, Belgium; Mastermix QuantiNova Probe RT-PCR Kit from Qiagen). Absolute quantification was done by a standard curve, based on a diluted PCR product. The cut-off value for normalisation is defined by the 95% upper CI limit for healthy controls.

## Statistics

Samples with readings above the standard curves were set at limit (2000 mg/kg for Calpro and 3000 mg/kg for Calprest). Day-to-day variation and head-to-head comparison was performed by Pearson correlation and Bland-Altman limits of agreement analysis on raw data and ln-transformed values. For single predictor, Receiver operating characteristic analysis (ROC), the raw data was used. A combined predictor of [MES + calprotectin] was constructed. To give the two base variables equal weight, both were standardized: [MES+calprotectin] = [MES/max MES] + [ln(calprotectin)/ln(max calprotectin*)]; for each assay (*defined by the standard curve for each assay). These constructed variables span values from 0 to 2. Optimal cut-off was defined by maximal Youden J. Statistical analysis was performed in IBM SPSS Statistics for Windows, Version 25.0. Armonk, NY, USA. Plots were produced in GraphPad Prism version 7.05 for Windows (GraphPad Software, La Jolla, CA, USA).

## Results

### Inter-assay variation

51 dual samples were extracted and analysed on both assays. Samples were collected within 1 week, but not on the same day. Both assays showed a fair correlation between the two samples with Pearson r = 0.744; P < 0.0001 for Calprest, and Pearson r = 0.729 P < 0.0001 for Calpro

(logarithmic transformation applied, please see Fig 2A and 2B). Bland-Altman plots showed that the variance of both assays were evenly distributed with no clinically relevant bias and a fairly equal limit of agreement (Fig 2C and 2D). These variations cover the total variance from sampling on different days, sample handling, extraction, and the ELISA method itself.

## Method comparison

For direct comparison of the two assays, 186 samples from different individuals were extracted and the same extract was run using both ELISA assays. A direct correlation plot shows a

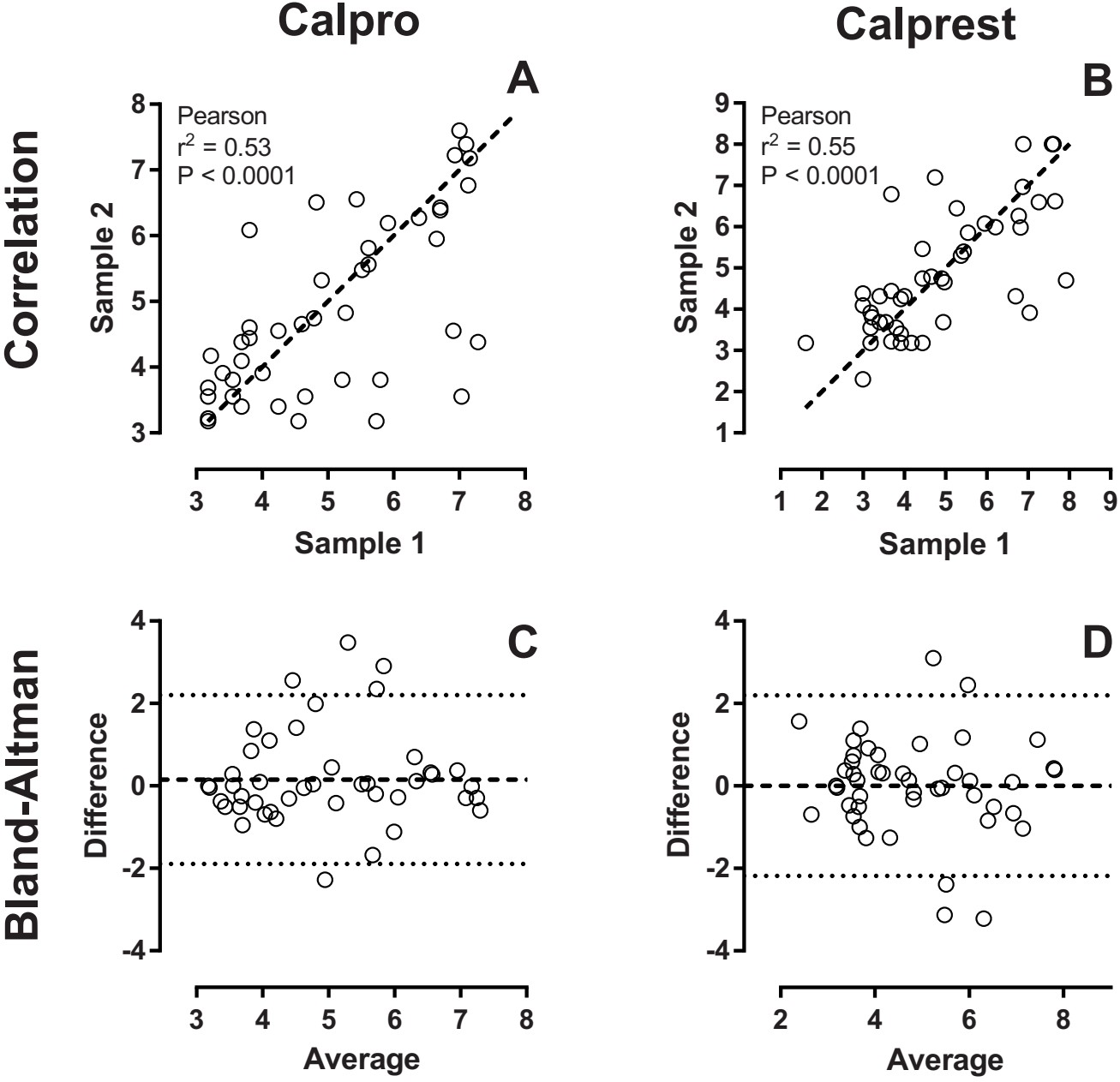

**Fig 2. Day-to-day variation.** Calprotectin was measured in 51 sets of 2 fecal samples collected within 1 week from each participant. Samples were extracted and run on both calprotectin assays. Values were then ln transformed. Panels A and B show scatterplots of the two samples for both assays. Dashed line is *line of identity*. Panels C and D show Bland-Altman plots of the same samples.

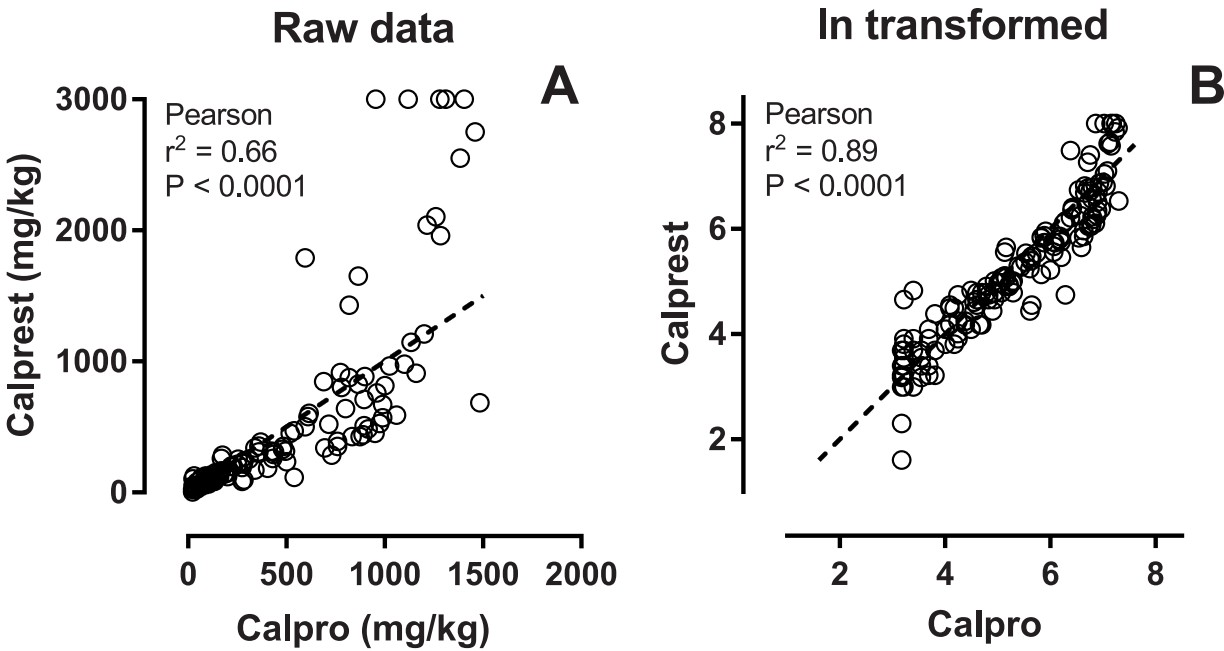

**Fig 3. Method comparison.** One-hundred-eightysix samples measured with both assays. Panel A shows the raw data. Below 1000 there is a fairly linear correlation, but in the higher range the Calprest assay measures higher values than Calpro. After ln transformation of the raw data, a significant linear trend can be seen in panel B. Dashed line is *line of identity*.

systematic skewing towards higher measurements in the high end using the Calprest kit. This reflects that the standard curve for Calprest extends to 3000, while Calpro stops at 2000. Thus a higher dynamic for the most elevated calprotectin values is expected using the Calprest assay (Fig 3A). Using log-transformed values, a good correlation can be seen (Fig 3B).

A Bland-Altman plot shows a tendency of lower mid-range values for the Calprest compared to the Calpro values (Fig 4A and 4B). This plot also shows that the range can be divided into sections: values of 0–200; 201–1000; and above 1000 (Fig 4C). Mean bias between the methods can be calculated as follows: segment 0–200: Calprest measures 3.9% lower than Calpro (transformed mean bias -0.04); 201–1000: Calprest measures 30% lower than Calpro (transformed mean bias -0.36); above 1000: too few datapoints, and the difference in maximal scale value makes Bland Altman analysis difficult to interpret in this segment. Calprest measures values 60% higher than Calpro (transformed mean bias of 0.47).

### ROC analysis

For ROC analysis 104 observations were available for the MES, and 96 for the TNF gene expression measurement. A ROC curve for prediction of deep remission defined by MES of 0 is shown in Fig 5A; Both assays had a significant AUC: 0.73 (0.63–0.82; P<0.0005) for Calpro, and 0.73 (0.64–0.83; P<0.0005) for Calprest, respectively. The Calprest kit may be marginally better at predicting MES 0 (Youden J 0.45; cut-off 100; sensitivity 0.86; specificity 0.60) vs Calpro (Youden J 0.41; cut-off 98; sensitivity 0.83; specificity 0.58). For both methods, however, a relatively low specificity is noted, and using these cut-off values may yield a considerable part of endoscopies with MES 1. However, for higher specificity, a cut-off of 28 is recommended for both kits (Calprest: specificity 0.82, sensitivity 0.38; Calpro: specificity 0.77, sensitivity 0.50).

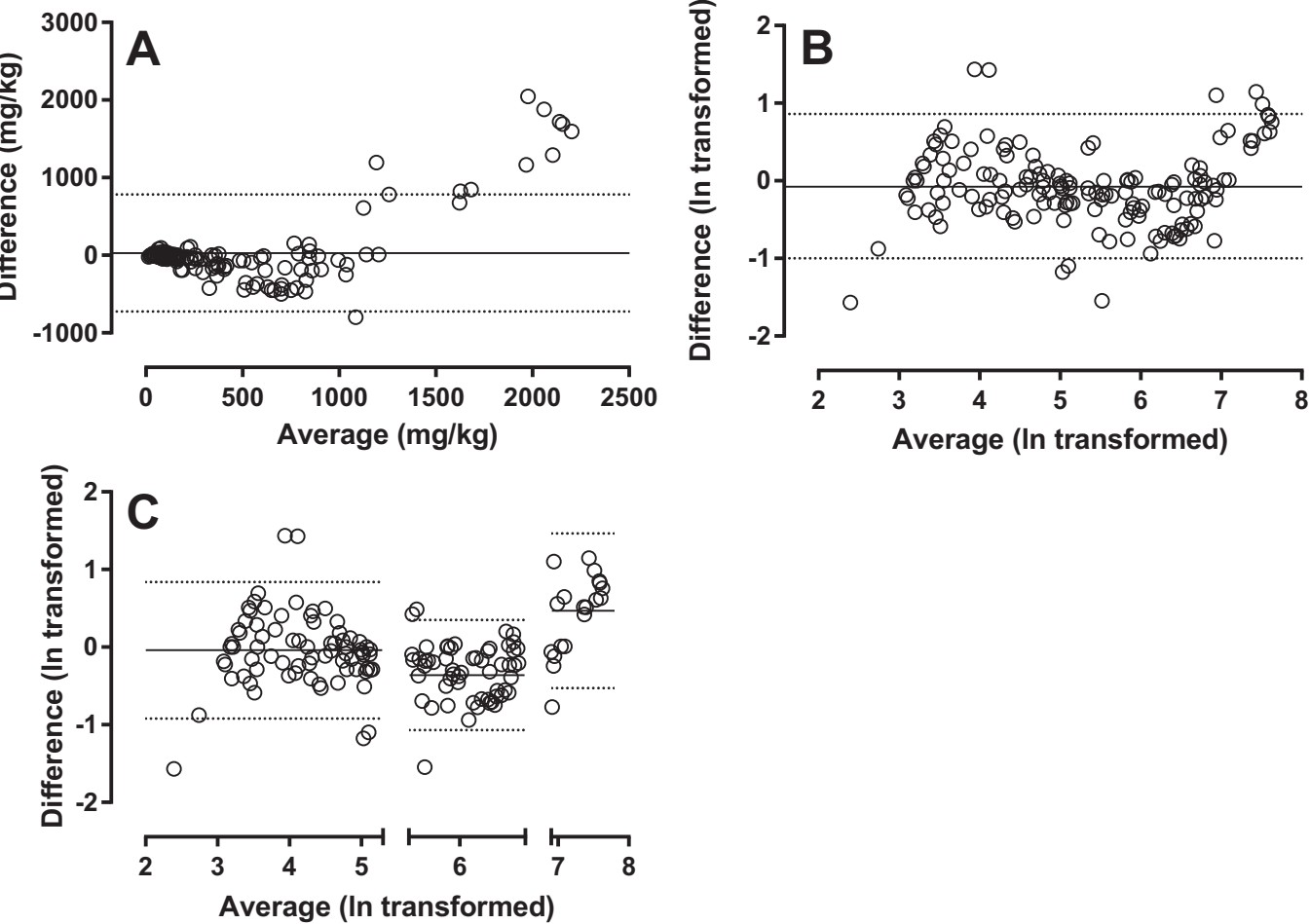

**Fig 4. Bland-Altman.** Analysis of 186 samples measured with both assays. The raw data in panel A shows shifting bias along the x-axis. This tendency is confirmed in the Bland-Altman plot of ln transformed raw data in panel B. Three sections with distinct bias pattern can be defined: calprotectin values <200; values in the interval 200–1000; and values >1000. Bland-Altman plots with mean bias and limits of agreement of each of these segments are presented in panel C.

A ROC curve for prediction of significant inflammation, defined by MES of 2–3, is shown in Fig 5B. Both assays again have significant AUC's: 0.79 (0.70–0.89; P<0.0005) for Calpro, and 0.80 (0.70–0.90; P<0.0005) for Calprest. Optimal cut-off by maximal Youden J: 138 (J = 0.48; sensitivity = 0.72; specificity = 0.76) for Calpro, and 73 (J = 0.53; sensitivity = 0.88; specificity = 0.65) for Calprest. Cut-off for specificity 0.80: 208 (J = 0.44; sensitivity = 0.64; specificity = 0.80) for Calpro and 148 (J = 0.52; sensitivity = 0.72; specificity = 0.80) for Calprest.

We also tested if mucosal TNF gene expression could be predicted by fecal calprotectin. Using earlier defined cut-off for normal TNF gene expression of <7500 copies/ug total RNA, the samples were categorized as normal or high. We then entered this as actual state variable in a ROC curve for both assays (Fig 5C). The curves did not yield significant AUC's: 0.58 (0.46–0.69; P = 0.24) for Calpro, and 0.57 (0.46–0.68; P = 0.29) for Calprest.

Finally, we tested a combination of MES and calprotectin (Fig 5D) For Calpro combined with MES, AUC was 0.72 (0.62–0.82; P = 0.001); for Calprest combined with MES the AUC was 0.71 (0.61–0.81; P = 0.001). Optimal cut-off by max Youden J was 0.81 for [MES+Calpro] (J = 0.43; sensitivity = 0.85; specificity = 0.58); and 0.79 for [MES+Calprest] (J = 0.46;

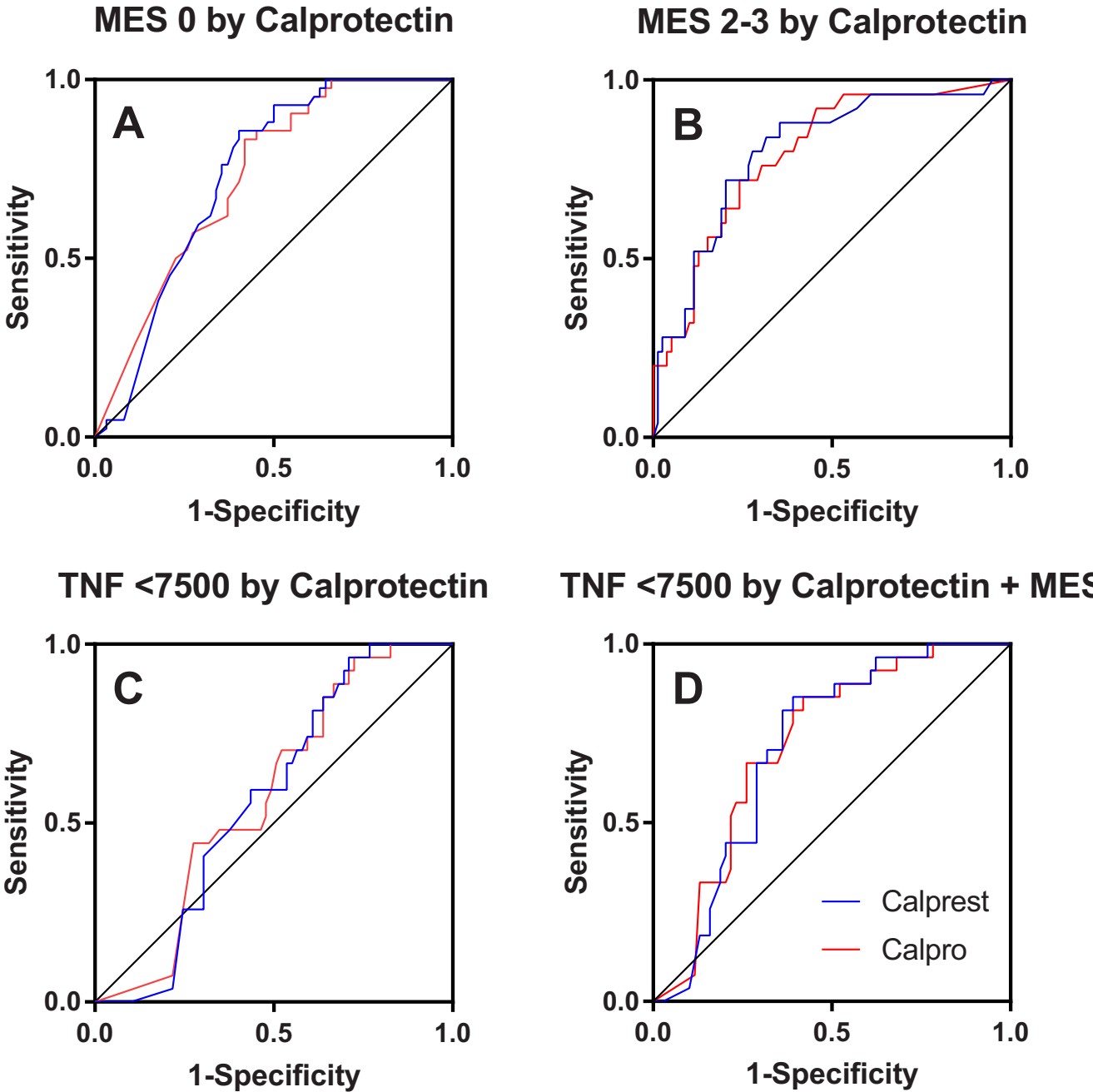

**Fig 5. ROC curves.** Prediction of endoscopic inflammatory activity. Panel A: Mucosal healing defined by Mayo endoscopic subscore (MES) = 0 by calprotectin (actual state: positives 42; negatives 62). Panel B: Significant inflammatory activity defined by MES 2–3 by calprotectin (actual state: positives 25; negatives 79). Panel C: Normalized mucosal TNF gene expression (<7500 copies/ug total RNA) by calprotectin (actual state: positives 27; negatives 69). Panel D: Normalized TNF gene expression (<7500 copies/ug total RNA) by the sum of calprotectin and MES (both standardized by ln transformation and division with the maximal scale value). For all plots: blue curve for Calprest; red curve for Calpro.

sensitivity = 0.85; specificity = 0.61). For specificity of 0.80, a cut-off of 0.51 for [MES+Calpro] (J = 0.13; sensitivity 0.33; specificity = 0.80); and a cut-off of 0.51 for [MES+Calprest] (J = 0.24; sensitivity 0.44; specificity 0.80) were found.

## Discussion

We have performed an inter-assay variation analysis and head-to-head comparison of two commercial calprotectin kits, and explored cut-off values for MES 0, MES 2–3, and mucosal TNF transcript–features that are important to assess in follow-up of UC patients. The kits both performed well but the optimal cut-off values varied somewhat, and must be determined for each manufacturer. Additionally, subtle differences were detected both in linearity and level, which probably rely on differences in standards and/or antibodies provided in the kits. It is possible to predict MES 0 to a certain extent, but calprotectin cannot yield both high sensitivity and specificity with the same cut-off. Eventually, calprotectin can be used to select patients for endoscopic evaluation to confirm MES 0. The calprotectin tests alone could not predict normalization of mucosal TNF expression, but by adding MES, the combined variables performed better in predicting mucosal TNF normalization, however still at quite low performances when using cut-off for high specificity (Youden 0.13 and 0.24 for Calpro and Calprest, respectively). Direct measurement of TNF expression still remains necessary for evaluation of immunological remission.

Day to day variation in calprotectin measurements is illustrated by the spread in Fig 2A and 2B. This is in agreement with earlier reports investigating repeated sampling and other factors, like time of day and different extraction devices [14,15]. The much lower error reported in similar studies is not comparable since the same samples were measured twice, and thus only reflect technical intra-assay error [16,17]. Since fecal samples mostly are collected by the patient, factors regarding collection are difficult to standardize. Serial samples (at least 2) will increase reliability. In our practice, two consecutive normal calprotectin measurements prompt endoscopy to ensure full remission. This approach saves endoscopy resources while still enabling close follow-up.

Comparison between values obtained from same samples on the two kits revealed a non-linear relationship. At values below 200 mg/kg, no significant systematic bias was detected, but mid-range values 200–1000 mg/kg are on average 30% lower in the Calprest assay compared to the Calpro assay. The lack of standardization prevent evaluation of which is more "right" [2]. The higher values measured by Calprest in samples above 1000 mg/kg is probably a consequence of a higher dynamic range for Caprest because of assay and standard curve differences. Compared to similar studies, the differences between Calpro and Calprest are relatively minute [17–19]. In comparison the Calpro assay performed quite similar but slightly superior to the HK325 (Hycultbiotech, Netherlands), and the EK-CAL ELISA (Bühlmann Laboratories AG, Switzerland) [20]

Our suggested cut-off for MES 0 is comparable to the findings of Kristensen et al [21] but considerably lower than that suggested for other assays [10] which underlines the necessity of kit-specific cut-off level. Interestingly, our earlier publication using mean of two samples yielded a better AUC for MES 0, and slightly different cut-off values for the Calpro kit [12]; this likely reflect that dual sampling yield better precision in a system with high day to day variation.

Detection of significant inflammation (MES 2–3), on the other hand, shows different cut-off levels reflecting the difference in detected values by the assays. Moreover, the difference in standard curves especially affect the mid-high range samples which show higher dynamic in the Calprest assay. This underlines the necessity of kit-specific cut-off values. A similar conclusion must be drawn from similar studies demonstrating highly different cut-off values [17,18,22]

The level of mucosal TNF transcript was poorly predicted by calprotectin values. It has been shown earlier that even in endoscopically normalised mucosa, not all patients have

normalised TNF transcript level [8]. Combined with MES, however, the ROC analysis showed improvement, though not at a level that can stand alone as evaluation of immunological remission. Most likely, this represents a deeper level of remission, and, so far, the only way to assess this feature is by measuring TNF directly in biopsies.

Weaknesses of the study are primarily that the endoscopic evaluation was not done by central reader, but graded by different clinicians. Moreover, a relatively low number of samples were used in the inter-assay evaluation (n = 51); however, the plots illustrate the high day to day variation, and the importance of repeated samples still stands. Regarding the ROC curves, the number of observations are reasonable, however not high enough to validate the proposed cut-off values by e.g. split-half analysis. The main point being that cut-off values are kit-specific still stands, though.

## Conclusions

In spite of a high day-today variation, calprotectin remains one of the best non-invasive measures of the inflammatory state in ulcerative colitis. However, even between these two highly correlated ELISA kits for calprotectin there are important differences, also in the interval containing cut-off values for clinical decisions. Consequently, due to the lack of standardisation for calprotectin measurement, cut-off values should be regarded as kit-specific. Both kits were most precise in predicting active inflammation (MES 2–3), but less so for prediction of mucosal healing (MES 0) and normalization of mucosal TNF gene expression. The Calprest assay has a higher dynamic range, the use of which is still to be determined. Calprotectin alone could not predict mucosal TNF transcript normalization.

## Supporting information

**S1 File. Raw data and metadata is available in SPSS format.**
(SAV)

## Acknowledgments

We thank Marian Remijn for expert technical assistance.

## Author Contributions

**Conceptualization:** Rasmus Goll, Katrine Carlsen, Jon Florholmen.

**Data curation:** Rasmus Goll.

**Formal analysis:** Rasmus Goll, Katrine Carlsen.

**Resources:** Rasmus Goll, Richard Heitmann, Øystein Kittel Moe, Katrine Carlsen, Jon Florholmen.

**Visualization:** Rasmus Goll.

**Writing – original draft:** Rasmus Goll, Richard Heitmann.

**Writing – review & editing:** Richard Heitmann, Øystein Kittel Moe, Katrine Carlsen, Jon Florholmen.

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
