## [Decision Letter · Decision Letter 0]

24 Jul 2019

PONE-D-19-17736

Head to head comparison of two commercial fecal calprotectin kits as predictor of Mayo endoscopic subscore and mucosal TNF expression in ulcerative colitis

PLOS ONE

Dear Dr Goll,

Thank you for submitting your manuscript to PLOS ONE. After careful consideration, we feel that it has merit but does not fully meet PLOS ONE’s publication criteria as it currently stands. Therefore, we invite you to submit a revised version of the manuscript that addresses the points raised during the review process.

Standardization is a frequent problem with assays such as fecal calprotectin. Diagnostic cut-off determined with one assay do not necessary apply for another assay.

It is useful to compare different producer’s tests and I like the attempt to use an independent parameter as TNF transcripts. Unfortunate the authors only compare two of the many available assays for fecal calprotectin, extending to more would have increased the usefulness of the paper.

I wonder if the authors have thought about directly comparing the assays with spiked samples? Using a given amount of WBC and stool from non-human species could have provided exact limit of detection/quantification and could have normalized the assays.

It seem rather unclear how many samples/patients that have been used in the present paper, I would suggest providing a flowchart.

Have the authors got permission to use data from the authors and journal of the previous paper (Scand J Gastroenterol. 260 2018 Jul 3;53(7):825–30).

We would appreciate receiving your revised manuscript by Sep 07 2019 11:59PM. To enhance the reproducibility of your results, we recommend that if applicable you deposit your laboratory protocols in protocols.io, where a protocol can be assigned its own identifier (DOI) such that it can be cited independently in the future. For instructions see: http://journals.plos.org/plosone/s/submission-guidelines#loc-laboratory-protocols

We look forward to receiving your revised manuscript.

Kind regards,

Pal Bela Szecsi, M.D. D.M.Sci.

Academic Editor

PLOS ONE

Journal Requirements:

Reviewers' comments:

Reviewer's Responses to Questions

**Comments to the Author**

1. Is the manuscript technically sound, and do the data support the conclusions?

Reviewer #1: Yes

Reviewer #2: Partly

2. Has the statistical analysis been performed appropriately and rigorously? 

Reviewer #1: Yes

Reviewer #2: Yes

3. Have the authors made all data underlying the findings in their manuscript fully available?

Reviewer #1: No

Reviewer #2: Yes

4. Is the manuscript presented in an intelligible fashion and written in standard English?

Reviewer #1: Yes

Reviewer #2: Yes

5. Review Comments to the Author

Reviewer #1: Regarding the manuscript by Goll et al.

Today F-calprotectin assays are widely used and is an important part of the workup of patients with suspected IBD. Evaluation assays and the correlation with TNF expression are thus important.

I think the data is clearly presented and I have only minor comments.

Abstract line 25: I am not completely enthusiastic about the use of calibration in this context. Calibration is more often used for assay calibrations and here the authors are relating the values to clinical states which is much more complex and also more valuable. I believe that the word calibration here may be misinterpreted by some readers. I thus suggest that the authors try and change calibration to another word.

Page 3, line 45: To some extent, part of… Please remove either as this is a duplication

Page 3 line 56. Add a space before (9-11)

Page 4: The extraction process: How similar were the sample extractions. Same sample? Were the samples mixed before sampling?

Page 5 and 6 method comparison: Most likely the authors had results that were above and below the measuring ranges of the assays. How were such samples handled? Values set at the limits? Analyzed rediluted? Omitted?

Page 7 line 164: Add a space between andhead-to

Table 1: Please add a table legend that also should include the units for the F-calprotectin assays.

Figure 3: please add the units

Please a sentence about availability of data. The PLOS Data policy requires authors to make all data underlying the findings described in their manuscript fully available without restriction, with rare exception (please refer to the Data Availability Statement in the manuscript PDF file). The data should be provided as part of the manuscript or its supporting information, or deposited to a public repository

Reviewer #2: Material and methods

When the authors refer to the two Calprptectin kits, they have to specify which methodology they use, and they put it in results, at the end of Inter-assay variation.

Nor do they indicate the cut-off from which they consider the TNF activity of the biopsies to be positive.

The total number of samples used for the comparative study is not understood, as they indicate that data from another study have been used, but everything is very unclear. In addition, the controls for which they were used are not well known, as no results are given either in Table 1.

Results

The first fragment of results should actually be included in Material and methods (lines 102 to 112), because there are not results on it.

I do not understand how lowering the cut-off increases specificity and lowers sensitivity (lines 144 and 145).

Discussion

The results have been compared with few previous results, hence the scarce bibliography.

Table 1

The legends of the abbreviations are missing.

Errata

Some errors should be corrected, as in Line 60: Calpro (Calpro Norway), and some words together, without spaces.

6. PLOS authors have the option to publish the peer review history of their article (what does this mean?). If published, this will include your full peer review and any attached files.

Reviewer #1: No

Reviewer #2: No

---

## [Author Response · Author response to Decision Letter 0]

28 Aug 2019

PONE-D-19-17736

Head to head comparison of two commercial fecal calprotectin kits as predictor of Mayo endoscopic subscore and mucosal TNF expression in ulcerative colitis

PLOS ONE

Dear Dr Goll,

Thank you for submitting your manuscript to PLOS ONE. After careful consideration, we feel that it has merit but does not fully meet PLOS ONE’s publication criteria as it currently stands. Therefore, we invite you to submit a revised version of the manuscript that addresses the points raised during the review process.

Standardization is a frequent problem with assays such as fecal calprotectin. Diagnostic cut-off determined with one assay do not necessary apply for another assay.

It is useful to compare different producer’s tests and I like the attempt to use an independent parameter as TNF transcripts. Unfortunate the authors only compare two of the many available assays for fecal calprotectin, extending to more would have increased the usefulness of the paper.

Response: Although we agree on this point, only a limited number of the samples are available as of today, and rerunning on other assays would reduce the number of samples in the analysis substantially. 

I wonder if the authors have thought about directly comparing the assays with spiked samples? Using a given amount of WBC and stool from non-human species could have provided exact limit of detection/quantification and could have normalized the assays.

Response: Being of technical interest, such experiments would have limited use for the clinical interpretation of measured values. Besides, this would be the type of validation already performed by the company’s marketing the tests. 

It seem rather unclear how many samples/patients that have been used in the present paper, I would suggest providing a flowchart.

Response: Thanks for the suggestion. We have provided a flowchart figure and tried to clean up the description of patients and samples.

Have the authors got permission to use data from the authors and journal of the previous paper (Scand J Gastroenterol. 260 2018 Jul 3;53(7):825–30).

Response: As can be seen in the flow chart figure, all samples in the present study were collected in Tromsø by the group of authors for the present paper. 

We would appreciate receiving your revised manuscript by Sep 07 2019 11:59PM. To enhance the reproducibility of your results, we recommend that if applicable you deposit your laboratory protocols in protocols.io, where a protocol can be assigned its own identifier (DOI) such that it can be cited independently in the future. For instructions see: http://journals.plos.org/plosone/s/submission-guidelines#loc-laboratory-protocols

• A rebuttal letter that responds to each point raised by the academic editor and reviewer(s). This letter should be uploaded as separate file and labeled 'Response to Reviewers'.

• A marked-up copy of your manuscript that highlights changes made to the original version. This file should be uploaded as separate file and labeled 'Revised Manuscript with Track Changes'.

• An unmarked version of your revised paper without tracked changes. This file should be uploaded as separate file and labeled 'Manuscript'.

We look forward to receiving your revised manuscript.

Kind regards,

Pal Bela Szecsi, M.D. D.M.Sci.

Academic Editor

PLOS ONE

Journal Requirements:

Reviewers' comments:

Reviewer's Responses to Questions

Comments to the Author

1. Is the manuscript technically sound, and do the data support the conclusions?

Reviewer #1: Yes

Reviewer #2: Partly

2. Has the statistical analysis been performed appropriately and rigorously? 

Reviewer #1: Yes

Reviewer #2: Yes

3. Have the authors made all data underlying the findings in their manuscript fully available?

Reviewer #1: No

Reviewer #2: Yes

Response: The raw data was submited in a supplementary file

4. Is the manuscript presented in an intelligible fashion and written in standard English?

Reviewer #1: Yes

Reviewer #2: Yes

5. Review Comments to the Author

Reviewer #1: Regarding the manuscript by Goll et al.

Today F-calprotectin assays are widely used and is an important part of the workup of patients with suspected IBD. Evaluation assays and the correlation with TNF expression are thus important.

I think the data is clearly presented and I have only minor comments.

Abstract line 25: I am not completely enthusiastic about the use of calibration in this context. Calibration is more often used for assay calibrations and here the authors are relating the values to clinical states which is much more complex and also more valuable. I believe that the word calibration here may be misinterpreted by some readers. I thus suggest that the authors try and change calibration to another word.

Response: In biomarker research, calibration is more widely used as compared to the more technical usage in assay design. However, we have rephrased from “calibration” to “tuning” throughout the manuscript.

Page 3, line 45: To some extent, part of… Please remove either as this is a duplication

Response: Point taken

Page 3 line 56. Add a space before (9-11)

Response: Thank you

Page 4: The extraction process: How similar were the sample extractions. Same sample? 

Response: The extraction devices provided with these two kits are very similar, while differences in extraction buffer may exist. 

Were the samples mixed before sampling? 

Response: Yes, the samples were mixed

We have rephrased the description on p 5 in the revised manuscript

Page 5 and 6 method comparison: Most likely the authors had results that were above and below the measuring ranges of the assays. How were such samples handled? Values set at the limits? Analyzed rediluted? Omitted? 

Response: For low/high samples, values were set at the limits; we added this in the revised manuscript in line 102

Page 7 line 164: Add a space between andhead-to

Response: Thank you

Table 1: Please add a table legend that also should include the units for the F-calprotectin assays.

Response: We have added this to the legend. 

Figure 3: please add the units

Response: We have added this to the figure

Please a sentence about availability of data. The PLOS Data policy requires authors to make all data underlying the findings described in their manuscript fully available without restriction, with rare exception (please refer to the Data Availability Statement in the manuscript PDF file). The data should be provided as part of the manuscript or its supporting information, or deposited to a public repository

Response: We added a sentence in the remarks at the end of the manuscript. Line 224

Reviewer #2: Material and methods

When the authors refer to the two Calprptectin kits, they have to specify which methodology they use, and they put it in results, at the end of Inter-assay variation.

Response: Duly noted. We have added methodology to “aim” in line 63 as well as in “methods” line 78

Nor do they indicate the cut-off from which they consider the TNF activity of the biopsies to be positive.

Response: Thanks for this comment. The assay is in-house, rendering the exact cutoff value of less interest for the readers. However, it is defined as upper 95% CI of healthy controls. In our lab the cut-off is 7500 copies/ug total RNA in extracts from colon mucosa – this is already stated in the ROC section of results; we have now clarified the method section in this regard. Line 99

The total number of samples used for the comparative study is not understood, as they indicate that data from another study have been used, but everything is very unclear. 

Response: Point taken, we have added a flow diagram to clarify

In addition, the controls for which they were used are not well known, as no results are given either in Table 1. 

Response: The 16 normal controls were entered only in the technical evaluations (day-to-day variation and assay comparison). For a technical evaluation, the source of fecal matter is less important. 

Results

The first fragment of results should actually be included in Material and methods (lines 102 to 112), because there are not results on it.

Response: This can be disputed, but we have moved the paragraph to the methods section and cleaned up a bit to clarify the sample overview.

I do not understand how lowering the cut-off increases specificity and lowers sensitivity (lines 144 and 145).

Response: This depends on the direction of the predictor variable and the target variable. In the final ROC analysis a lower MES/calpro value yields higher probability of normal TNF. Thus lowering cut-off will increase specificity and lower sensitivity. 

Discussion

The results have been compared with few previous results, hence the scarce bibliography.

Response: We have expanded the discussion a bit, citing 5 more earlier works. 

Table 1

The legends of the abbreviations are missing.

Response: This has been added 

Errata

Some errors should be corrected, as in Line 60: Calpro (Calpro Norway), and some words together, without spaces.

Response: Thank you; these errors have been corrected

6. PLOS authors have the option to publish the peer review history of their article (what does this mean?). If published, this will include your full peer review and any attached files.

Do you want your identity to be public for this peer review? For information about this choice, including consent withdrawal, please see ourPrivacy Policy.

Reviewer #1: No

Reviewer #2: No

While revising your submission, please upload your figure files to the Preflight Analysis and Conversion Engine (PACE) digital diagnostic tool,https://pacev2.apexcovantage.com/. PACE helps ensure that figures meet PLOS requirements. To use PACE, you must first register as a user. Registration is free. Then, login and navigate to the UPLOAD tab, where you will find detailed instructions on how to use the tool. If you encounter any issues or have any questions when using PACE, please email us at figures@plos.org. Please note that Supporting Information files do not need this step.

---

## [Decision Letter · Decision Letter 1]

17 Sep 2019

PONE-D-19-17736R1

Head to head comparison of two commercial fecal calprotectin kits as predictor of Mayo endoscopic subscore and mucosal TNF expression in ulcerative colitis

PLOS ONE

Dear Dr Goll,

Thank you for submitting your manuscript to PLOS ONE. After careful consideration, we feel that it has merit but does not fully meet PLOS ONE’s publication criteria as it currently stands. Therefore, we invite you to submit a revised version of the manuscript that addresses the points raised during the review process.

The manuscript have greatly improved, but some minor issues still remain.

Please address the point reviewer #2 has raised.

We would appreciate receiving your revised manuscript by Nov 01 2019 11:59PM. To enhance the reproducibility of your results, we recommend that if applicable you deposit your laboratory protocols in protocols.io, where a protocol can be assigned its own identifier (DOI) such that it can be cited independently in the future. For instructions see: http://journals.plos.org/plosone/s/submission-guidelines#loc-laboratory-protocols

We look forward to receiving your revised manuscript.

Kind regards,

Pal Bela Szecsi, M.D. D.M.Sci.

Academic Editor

PLOS ONE

Reviewers' comments:

Reviewer's Responses to Questions

**Comments to the Author**

1. If the authors have adequately addressed your comments raised in a previous round of review and you feel that this manuscript is now acceptable for publication, you may indicate that here to bypass the “Comments to the Author” section, enter your conflict of interest statement in the “Confidential to Editor” section, and submit your "Accept" recommendation.

Reviewer #1: All comments have been addressed

Reviewer #2: (No Response)

2. Is the manuscript technically sound, and do the data support the conclusions?

Reviewer #1: (No Response)

Reviewer #2: Partly

3. Has the statistical analysis been performed appropriately and rigorously? 

Reviewer #1: (No Response)

Reviewer #2: Yes

4. Have the authors made all data underlying the findings in their manuscript fully available?

Reviewer #1: (No Response)

Reviewer #2: Yes

5. Is the manuscript presented in an intelligible fashion and written in standard English?

Reviewer #1: (No Response)

Reviewer #2: Yes

6. Review Comments to the Author

Reviewer #1: (No Response)

Reviewer #2: Abstract

- Line 35: In the sentence "and at higher values Calprest measured higher values than Calpro", the authors should indicate the average increase increase of Calprest over Calpro in % at values >1000 mg/Kg.

- Lines 37 and 38: the authors use the terms "reasonable precision" and "transcript values reasonably well" subjectively, rather than objective identifiers in quantitative terms.

- Line 58: the authors continue to use the term 'calibrating'. Instead, it would be more appropriate to use the terms "adjustment" or "setting", better than tuning.

Material and methods

- Line 66: delete "Patients with UC were recruited first" because it is repeated later; add "as part of a prospective project at the Department of Gastroenterology, University hospital of North Norway" behind (12), on line 70.

- Line 97: add manufacturer and locality of TaqMan assays for TNF.

- Line 11: GraphPad Software, La Jolla, CA, USA, best put in parentheses.

Results

- Lines 145-146: I disagree that when the cut-off goes down, the sensitivity drops and the specificity increases. When lowering the cut-off, more cases of UC are diagnosed (sensitivity increases), but also other non-inflammatory diseases, with a greater number of false positives (specificity decreases). These data should be reviewed.

I do not understand how you have obtained these results; it is not understandable or reasonable. In fact, to increase specificity, the cut-off increase above 100 later (lines 150-151).

- Line 146: correct "maksimal".

- Line 153: separate "totalRNA".

- Lines 160 and 161: it is noteworthy that the Youden indices obtained with the MES + Calprotectin in both trials are very low (0.13 and 0.24), and the authors do not refer to it in the Discussion.

Table I: The result of the 16 controls has not been included in the trial comparison.

Figure 2: The authors should add the line of the line and refer in the text that the correlation between the calprotectin results of two samples from the same patient is weak by both methods (Pearson's r of 0.53 and 0.55).

Figure 5: the authors must add in the ROC curves the kit that corresponds to each color (blue and red), and also in the legend, in brackets TNF <7500 after “Normalized mucosal TNF gene expression”.

Conclusion

Add some mention to the prediction of inflammation or its cure (MES), as you indicated in the summary.

References

- Reference 11 (line 260), change 1936.e1. for 36.e1.

- Reference 20: the year is incorrect; volume and pages are missing and the authors are incompletely cited.

7. PLOS authors have the option to publish the peer review history of their article (what does this mean?). If published, this will include your full peer review and any attached files.

Reviewer #1: No

Reviewer #2: No

---

## [Author Response · Author response to Decision Letter 1]

9 Oct 2019

please see attached file "response to reviewers"

---

## [Decision Letter · Decision Letter 2]

15 Oct 2019

PONE-D-19-17736R2

Head to head comparison of two commercial fecal calprotectin kits as predictor of Mayo endoscopic sub-score and mucosal TNF expression in ulcerative colitis

PLOS ONE

Dear Dr Goll,

Thank you for submitting your manuscript to PLOS ONE. After careful consideration, we feel that it has merit but does not fully meet PLOS ONE’s publication criteria as it currently stands. Therefore, we invite you to submit a revised version of the manuscript that addresses the points raised during the review process.

Some minor issues still needs your attention, please correct.

We would appreciate receiving your revised manuscript by Nov 29 2019 11:59PM. To enhance the reproducibility of your results, we recommend that if applicable you deposit your laboratory protocols in protocols.io, where a protocol can be assigned its own identifier (DOI) such that it can be cited independently in the future. For instructions see: http://journals.plos.org/plosone/s/submission-guidelines#loc-laboratory-protocols

We look forward to receiving your revised manuscript.

Kind regards,

Pal Bela Szecsi, M.D. D.M.Sci.

Academic Editor

PLOS ONE

Reviewers' comments:

Reviewer's Responses to Questions

**Comments to the Author**

1. If the authors have adequately addressed your comments raised in a previous round of review and you feel that this manuscript is now acceptable for publication, you may indicate that here to bypass the “Comments to the Author” section, enter your conflict of interest statement in the “Confidential to Editor” section, and submit your "Accept" recommendation.

Reviewer #1: All comments have been addressed

Reviewer #2: (No Response)

2. Is the manuscript technically sound, and do the data support the conclusions?

Reviewer #1: (No Response)

Reviewer #2: Yes

3. Has the statistical analysis been performed appropriately and rigorously? 

Reviewer #1: (No Response)

Reviewer #2: Yes

4. Have the authors made all data underlying the findings in their manuscript fully available?

Reviewer #1: (No Response)

Reviewer #2: Yes

5. Is the manuscript presented in an intelligible fashion and written in standard English?

Reviewer #1: (No Response)

Reviewer #2: Yes

6. Review Comments to the Author

Reviewer #1: (No Response)

Reviewer #2: The authors have greatly improved the manuscript quality, there are only minor details to modify before publication. These are the following:

Authors' contributions: delete Summary (line 17).

Keywords: delete "calibration".

Lines 145-145. Now I understand the objective of the authors: different cut-off points according to the MES searched (positive/negative). That is correct, but clinically this is difficult to establish. Laboratory tests usually have a single cut-off point, and results that are above/below are considered positive/negative (or vice versa). Otherwise, a grey area of results is established, difficult to classify.

I accept your explanation.

Results

Add a full stop, new paragrah behind of"...for Calprest." (line 160).

Conclusions

Add to the end of the Conclusions the statement that the authors include in the Abstract, but that does not appear in the manuscript text (already indicated in the previous review): "Both kits were more accurate in the prediction of active inflammation (MES 2-3), but less for the prediction of mucosal healing (MES 0) and normalization of the expression of the mucosal TNF gene ".

Figure 2: I was referring to the straight line of identity, which you have already added by the authors.

7. PLOS authors have the option to publish the peer review history of their article (what does this mean?). If published, this will include your full peer review and any attached files.

Reviewer #1: No

Reviewer #2: No

---

## [Author Response · Author response to Decision Letter 2]

22 Oct 2019

Please see response in the file attached "response to reviewers round 3"

---

## [Editor Report · Decision Letter 3]

24 Oct 2019

Head to head comparison of two commercial fecal calprotectin kits as predictor of Mayo endoscopic sub-score and mucosal TNF expression in ulcerative colitis

PONE-D-19-17736R3

Dear Dr. Goll,

We are pleased to inform you that your manuscript has been judged scientifically suitable for publication and will be formally accepted for publication once it complies with all outstanding technical requirements.

With kind regards,

Pal Bela Szecsi, M.D. D.M.Sci.

Academic Editor

PLOS ONE
---

## [Editor Report · Acceptance letter]

15 Nov 2019

PONE-D-19-17736R3 

Head to head comparison of two commercial fecal calprotectin kits as predictor of Mayo endoscopic sub-score and mucosal TNF expression in ulcerative colitis 

Dear Dr. Goll:

I am pleased to inform you that your manuscript has been deemed suitable for publication in PLOS ONE. Congratulations! Your manuscript is now with our production department. 

With kind regards,

on behalf of

Dr. Pal Bela Szecsi 

Academic Editor

PLOS ONE